# N Multipliers for N Bits: Learning Bit Multipliers for Non-Uniform Quantization

**Raghav Singhal**  **Anmol Biswas**[†]  **Sivakumar Elangovan**  **Shreyas Sabnis**

**Udayan Ganguly**

Indian Institute of Technology Bombay, India
anmolbiswas@gmail.com[†]

## Abstract

Effective resource management is critical for deploying Deep Neural Networks (DNNs) in resource-constrained environments, highlighting the importance of low-bit quantization to optimize memory and speed. In this paper, we introduce N-Multipliers-for-N-Bits, a novel method for non-uniform quantization designed for efficient hardware implementation. Our method uses N parameters, distinct for every layer and corresponding to the N quantization bits, whose linear combinations span the set of allowed weights (and activations). Furthermore, we learn these parameters in conjunction with the weights, ensuring exceptional flexibility in the quantizer model with minimal hardware overhead. We validate our method on CIFAR-10 and ImageNet, achieving competitive results with 3- and 4-bit quantized models. We demonstrate strong performance on 4-bit quantized Spiking Neural Networks (SNNs), evaluated on the CIFAR10-DVS and N-Caltech 101 datasets. Further, we address the issue of stuck-at faults in hardware, and demonstrate robustness to up to 30% faulty bits.

## 1 Introduction

Deep learning dominates computer vision and broader AI applications, where cloud-based models perform inference by transferring data to servers. While effective, this approach is inefficient in terms of data transfer and power consumption. A more efficient alternative, especially for simple tasks, is edge inference using low-power accelerators with fixed-point arithmetic and in-memory or near-memory computing architectures [1–5]. These architectures, such as crossbar arrays, perform matrix-vector multiplication by accumulating parallel operations. They can be implemented using analog components or digital ones, but both approaches encounter a trade-off between energy efficiency and performance [6, 7]. Energy efficiency in edge devices often comes at the cost of circuit non-idealities such as line resistance and device variability [8–10] and hardware faults such as stuck-at (SA) faults [11, 12], where certain weight bits get stuck at either 0 or 1 and become unprogrammable. Addressing both quantization errors and hardware faults is crucial for optimizing edge inference.

Low-bit quantization for weights and activations has been extensively explored through quantization-aware training (QAT). Popular methods in the literature include regularization-based QAT, like sine-regularization [13, 14] and bin-regularization [15], and learned quantization scale [16–18]. Non-uniform methods, such as learning quantization levels or companding functions [19, 20], offer flexibility by learning key parameters. However, these works are not generally extendable to the specific challenges of low-power neuromorphic accelerators and event-driven Spiking Neural Networks (SNNs) [21, 22].

38th Second Workshop on Machine Learning with New Compute Paradigms at NeurIPS 2024(MLNCP 2024).

In this work, we propose a QAT scheme that optimizes bit-multipliers for each quantization level using a regularization-based approach using Mean Squared Error (MSE) loss. This enables the generalization of our method beyond just QAT for Artificial Neural Networks and into QAT for Spiking Neural Networks and non-ideality and hardware fault mitigation for low-power neuromorphic accelerators. We match the state-of-the-art performance in QAT for 4-bit networks on CIFAR-10 [23] and ImageNet [24] benchmarks and show excellent performance in 4-bit SNNs on CIFAR10-DVS [25] and N-Caltech 101 [26] neuromorphic benchmarks. Finally, we also show robust training for low-bit quantized models even with a high rate of hardware faults (up to $30\%$).

Our key contributions are summarized as follows:

- We introduce a novel, flexible, and hardware-compatible quantization framework that learns N bit multipliers per layer alongside network weights, enabling adaptable precision with minimal hardware overhead, while spanning a rich set of quantization levels.

- We show our method's effectiveness across multiple networks and datasets, achieving comparable state-of-the-art results for 3- and 4-bit DNNs on CIFAR-10 [23] and ImageNet [24], and 4-bit SNNs on event-based datasets: CIFAR10-DVS [25] and N-Caltech 101 [26].

- We propose a fault-tolerant quantization method that enables low-bit models to maintain performance up to $30\%$ faulty bits, as demonstrated on CIFAR-10, enhancing robustness.

- We propose a custom implementation of bit-level multipliers for analog/digital crossbars, optimized for our quantization scheme and directly portable to neuromorphic hardware.

## 2  Methodology

**Preliminaries:** Quantization aims to replace floating-point weights and activations in DNNs with low-bit representations to reduce memory usage and speed up computations. A general N-bit quantizer function will have $2^N$ levels, say $l_1, l_2, \ldots, l_{2^N}$, $2^N - 1$ transition thresholds, say $t_1, t_2, \ldots, t_{2^N-1}$, and is defined as follows:

$$Q(x) = \begin{cases} l_1 & \text{if } x < t_1 \\ l_i & \text{if } t_{i-1} \leq x < t_i, \quad i = 2, 3, \ldots, 2^N - 1 \\ l_{2^N} & \text{if } x \geq t_{2^N-1} \end{cases} \tag{1}$$

**Quantizer Model:** We introduce an $N$-dimensional learnable vector $r \in \mathbb{R}^N$, which defines the N bit multipliers, alongside a scalar offset $c$ in our quantizer model. The set of allowed quantized weights or activations is given by:

$$W_r = \left\{ \langle r, b \rangle + c \mid b \in \{0,1\}^N \right\} \tag{2}$$

The quantization function maps each full-precision weight to its nearest quantized counterpart:

$$\hat{x} = Q(x, r) = \arg \min_{w_q \in W_r} |x - w_q| \tag{3}$$

This design enables a flexible non-uniform quantizer with multiple step sizes, offering hardware efficiency while preserving the structure of N-bit quantization. Although learning all $2^N$ quantization levels would offer maximum flexibility, it would undermine hardware efficiency and the core benefits of N-bit quantization. Figure 1a illustrates a sample quantizer function. Drawing parallels between a general N-bit quantizer and the one introduced above, we can see that the elements of the set $W_r$ serve as the levels, $l_1, l_2, \ldots, l_{2^N}$, and the transition thresholds are defined as $t_i = (l_i + l_{i+1})/2$.

**Loss and Learning:** We jointly optimize the bit multipliers, offsets, and weights by introducing an additional quantization-aware loss alongside the standard cross-entropy loss. This allows the model parameters to be optimized through backpropagation within the usual training pipeline. During training, the weights remain in full precision but gradually align with their quantized counterparts due to the influence of the quantization-aware loss. The actual quantization is applied post-training, where the full-precision weights are mapped to their nearest quantized values.

**Quantization-Aware Loss:** We define a regularization loss that minimizes the squared error between each weight and its nearest quantized value. To balance gradient contributions across layers, we

introduce a layer-specific scaling factor. The total loss is formulated as:

$$\mathcal{L} = \mathcal{L}_{CE} + \lambda \sum_{l=1}^{L} \alpha_l \sum_{i=1}^{n_l} \min_{w_q \in W_r^l} | w_i - w_q |^2 \tag{4}$$

where $\mathcal{L}_{CE}$ is the cross-entropy loss and $W_r^l$ represents the set of quantized weights for layer $l$, defined by parameters $r^l$ and $c^l$. The term $\alpha_l$ is a layer-wise scaling factor, and $\lambda$ controls the regularization strength. Following other works[16], we set $\alpha_l$ as $1/\sqrt{N \cdot Q_P}$, where $Q_P$ is $2^b - 1$ for activations (unsigned data) and $2^{b-1} - 1$ for weights (signed data), respectively; $b$ denotes the number of bits. Figure 1b illustrates the regularization loss for a sample weight using an arbitrary vector $r$ to define the quantized weight set. Equivalently, the loss can be expressed as a function of the weights and bit multipliers. This formulation jointly optimizes the overall objective and the quantization parameters, including the bit multipliers and offsets that define the quantization function itself.

$$\mathcal{L} = \mathcal{L}_{CE} + \lambda \sum_{l=1}^{L} \alpha_l \sum_{i=1}^{n_l} | w_i - Q(w_i, r^l) |^2 \tag{5}$$

**Gradient Calculation:** The gradient calculation for the weights and quantizer parameters is

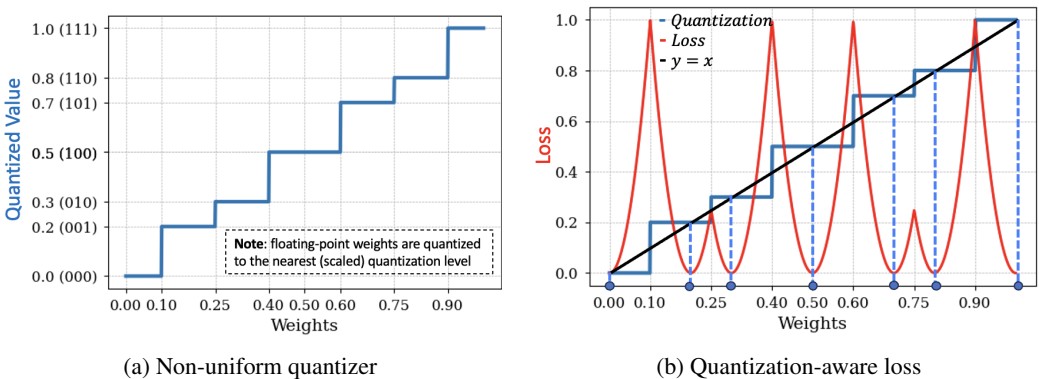

| (a) Non-uniform quantizer | (b) Quantization-aware loss |

Figure 1: **(a)** Non-uniform quantizer model demonstrating the learned bit-multiplier quantization function. **(b)** QAT loss with (MSE) regularization (shown in **red**) to minimize quantization error.

straightforward. Since we use full precision weights throughout the training, we can simply define $\frac{\partial Q(w,r)}{\partial w} = 0$, thereby eliminating the need of any gradient approximation techniques. For the quantizer parameters, $\frac{\partial Q(w,r)}{\partial c} = 1$ and $\nabla_r Q(w, r) = B_r(Q(w, r))$, where $B_r$ is an inverse map defined as $B_r : W_r \to \{0, 1\}^N$, providing the bit representation vector of the quantized weights. This encoding function satisfies $w_q = \langle r, B_r(w_q) \rangle + c \forall w_q \in W_r$. The gradients for the weights, bit multipliers, and offsets are calculated as follows:

$$\frac{\partial L}{\partial w} = \frac{\partial L_{CE}}{\partial w} + 2\lambda \cdot \alpha_l \cdot (w - Q(w, r^l)) \tag{6}$$

$$\frac{\partial L}{\partial r^l} = 2\lambda \cdot \alpha_l \sum_{i=1}^{n_l} (w_i - Q(w_i, r^l)) \cdot B_r(Q(w_i, r^l)) \tag{7}$$

$$\frac{\partial L}{\partial c^l} = 2\lambda \cdot \alpha_l \sum_{i=1}^{n_l} (w_i - Q(w_i, r^l)) \cdot (-\frac{\partial Q(w_i, r^l)}{\partial c^l}) = 2\lambda \cdot \alpha_l \sum_{i=1}^{n_l} (Q(w_i, r^l) - w_i) \tag{8}$$

**SNN Training:** SNNs inherently produce *quantized activations* in the form of *spike trains*, we thus need to solely quantize the weights of the network. We use a Leaky Integrate-and-Fire (LIF) model [27] for the spiking neuron in our SNN models. These discrete-time equations describe its dynamics:

$$H[t] = V[t-1] + \beta(X[t] - (V[t-1] - V_{reset})) \tag{9}$$
$$S[t] = \Theta(H[t] - V_{th}) \tag{10}$$
$$V[t] = H[t] \ (1 - S[t]) + V_{reset} \ S[t] \tag{11}$$

where $X[t]$ denotes the input current at time step $t$. $H[t]$ denotes the membrane potential following neural dynamics and $V[t]$ denotes the membrane potential after a spike at step $t$, respectively. The model uses a firing threshold $V_{th}$ and utilizes the Heaviside step function $\Theta(x)$ to determine spike generation. The output spike at step $t$ is denoted by $S[t]$, while $V_{reset}$ represents the reset potential following a spike. The membrane decay constant is denoted by $\beta$. To facilitate error backpropagation, we use the surrogate gradient method [28], defining $\Theta'(x) \triangleq \sigma'(x)$, where $\sigma(x)$ is the arctan surrogate function [29]. The remaining part of the training and quantization follows that of the non-spiking networks described earlier.

**Fault-Aware Modification:** We propose a two-pronged approach to address SA faults in quantized neural networks. Firstly, we enhance fault awareness during training by periodically (every 4 epochs) loading faulty weights onto the model. Secondly, we introduce a fault-aware modification to our algorithm, designed to avoid weight configurations rendered impossible by SA faults. We introduce a *validity* term that constrains weights to only those quantization levels that are achievable, avoiding those rendered unreachable by faulty bits. The *validity* term is defined for each layer as a binary map that indicates whether a specific weight can attain a given quantization level (1 if achievable, 0 otherwise). This allows us to modify the quantization-aware training loss in Equation 4 as follows:

$$\mathcal{L} = \mathcal{L}_{CE} + \lambda \sum_{l=1}^{L} \alpha_l \sum_{i=1}^{n_l} \min_{w_q \in W_r^l} (val_{i,q}^l \mid w_i - w_q \mid^2 + (1 - val_{i,q}^l) \cdot \Delta) \tag{12}$$

Here, $val_{i,q}^l$ represents the *validity* term for weight $w_i$ in layer $l$ with respect to the quantization level $w_q \in W_r^l$. If $w_i$ can reach $w_q$, then $val_{i,q}^l = 1$; otherwise, $val_{i,q}^l = 0$. The term $\Delta$ is a large constant that penalizes unreachable quantization levels, effectively excluding them from the optimization.

## 3   Experiments

We initialize quantized networks with weights from a trained full-precision model of the same architecture, then fine-tune in the quantized space, which hase been proven to improve performance [30–32]. We quantize input activations and weights to 3- or 4-bits for all matrix multiplication layers except the first and last, which use 8-bits. This approach is commonly used for quantizing deep networks, and has been proven to increase effectiveness at the cost of minimal overhead [16]. The weights and the quantization parameters: bit mutlipliers and the offset values, are trained using SGD with a momentum of 0.9 and a cosine learning rate decay schedule [33]. We sweep over different values of the regularization hyperparameter $\lambda$ and chose $\lambda = 100$ for our results.

**ANN Training Details.** We use the ResNet-18 [34] architecture for experiments on CIFAR-10 [23] and ImageNet [24] datasets. Models are trained for 200 epochs on CIFAR-10 and 90 epochs on ImageNet with the weights having a learning rate of 0.01 and 0.1 respectively. The other parameters are trained with a learning rate of 0.001. For ImageNet, we preprocess images by resizing them to $256 \times 256$ pixels. During training, we apply random $224 \times 224$ crops and horizontal flips half the time. At inference, we use a center crop of $224 \times 224$. For CIFAR-10, we augment the training data by padding images with 4 pixels on each side, then taking random 32x32 crops. We also apply random horizontal flips half the time. The results are shown in Table 1 and 2.

**SNN Training Details.** We use the ResNet-19 [35] and VGG-11 [36] models, after adapting them to SNNs. Specifically, we replace all ReLU activation functions with LIF modules and substitute max-pooling layers with average pooling operations. We follow the implementation and data augmentation technique used in NDA [37] as our baseline training method. The weights and the other parameters are trained with a learning rate of 0.01 and 0.001 respectively. We evaluate on the N-Caltech 101 and CIFAR10-DVS benchmarks. N-Caltech 101 consists of 8,831 DVS images converted from the original Caltech 101 dataset, while CIFAR10-DVS comprises 10,000 DVS images derived from the original CIFAR10 dataset. For both these datasets, we apply a 9:1 train-validation split and resize all images to $48 \times 48$. Each sample is temporally integrated into 10 frames using spikingjelly [38]. $V_{reset}$ is set to 0 and the membrane decay $\beta$ is 0.25. Our results are presented in Table 3.

**Fault-Aware Training.** We evaluate our method on the VGG-13 architecture, training with 3-bit and 4-bit precision for both weights and activations on the CIFAR-10 dataset. Our experiments consider varying levels of SA fault density. Figures 2a and 2b illustrate the efficacy of our approach for 4-bit and 3-bit quantization, respectively.

## 4 Results and Analysis

**Comparison with Baselines.** Tables 1 and 2 present our quantized ANN results for CIFAR10 and ImageNet, respectively. Our method outperforms existing approaches, with 4-bit ResNet-18 (**W4/A4** refers to 4-bit weights and 4-bit activations) achieving a $0.24\%$ accuracy increase over full-precision (FP) on CIFAR-10 and matching FP performance on ImageNet. For 4-bit quantized SNNs (Table 3), we observe performance gains on N-Caltech 101 and marginal losses on CIFAR10-DVS compared to FP. We attribute occasional performance improvements in both 4-bit ANNs and SNNs to the regularization effect of our quantization loss.

Table 1: Accuracy (%) for 3- and 4- bit quantized ResNet-18 models on CIFAR-10. FP denotes full-precision accuracy, $\Delta$ FP denotes difference in performance compared to the corresponding FP network. **Best**/second best relative performances for each bit-width are marked in **bold**/underlined.

| Method | FP | W4/A4 ($\Delta$ FP) | W3/A3 ($\Delta$ FP) |
|---|---|---|---|
| L1 Reg [39] | 93.54 | 89.98 ($-3.56$) | - |
| BASQ [40] | 91.7 | 90.21 ($-1.49$) | - |
| LTS [41] | 91.56 | 91.7 ($+0.1$) | 90.58 ($-0.98$) |
| PACT [17] | 91.7 | 91.3 ($-0.4$) | 91.1 ($-0.6$) |
| LQ-Nets [19] | 92.1 | - | 91.6 ($-0.5$) |
| LCQ [20] | 93.4 | 93.2 ($-0.2$) | 92.8 ($-0.6$) |
| **Ours (N-Multipliers)** | 93.26 | **93.50** ($+0.24$) | **92.84** ($-0.42$) |

Table 2: Accuracy (%) for 4-bit quantized ResNet-18 models on ImageNet. **Best**/second best performances are marked in **bold**/underlined.

| Method | Type | FP | W4/A4 ($\Delta$ FP) |
|---|---|---|---|
| L1 Reg [39] | No QAT | 69.7 | 57.5 ($-12.5$) |
| SinReQ [14] | Sine reg. | 70.5 | 64.6 ($-5.9$) |
| LTS [41] | Lottery | 69.6 | 68.3 ($-1.3$) |
| PACT [17] | Learned scale | 69.7 | 69.2 ($-0.5$) |
| LQ-Nets [19] | Linear non-uniform | 70.3 | 69.3 ($-1.0$) |
| QIL [42] | Non-linear | 70.2 | 70.1 ($-0.1$) |
| QSin [13] | Sine reg. | 69.8 | 69.7 ($-0.1$) |
| LCQ [20] | Non-linear | 70.4 | **71.5** ($+1.1$) |
| Ours | Fixed levels | 69.6 | 68.2 ($-1.4$) |
| Ours (N-Mult) | Linear non-uniform | 69.6 | 69.6 ($-0.0$) |

Table 3: Accuracy (%) for 4- bit quantized SNNs on CIFAR10-DVS and N-Caltech 101.

| Dataset | Model | FP | W4 ($\Delta$ FP) |
|---|---|---|---|
| CIFAR10-DVS | Spiking VGG-11 | 71.92 | 71.84 ($-0.08$) |
| CIFAR10-DVS | Spiking ResNet-19 | 72.91 | 72.14 ($-0.77$) |
| N-Caltech 101 | Spiking VGG-11 | 73.19 | 74.18 ($+0.99$) |
| N-Caltech 101 | Spiking ResNet-19 | 75.27 | 75.93 ($+0.66$) |

**Robustness to Faults.** SA faults represent extreme non-idealities in hardware, with each faulty bit halving the range of possible weight values. High device variability in conductance states can similarly cause significant discrepancies between expected and realized weights. Our approach, combining periodic loading of faulty weights during training with a fault-aware modified QAT algorithm, demonstrates robust performance even under high SA fault densities.

**Hardware Compatibility.** Figures 3a and 3b illustrate implementations of custom bit multipliers in analog and digital crossbar arrays. For analog arrays, the implementation incurs no additional cost, requiring only the adjustment of bit-multiplier conductance values from power-of-2 proportions to custom values. In digital arrays, the multiply-accumulate operation remains fully fixed-point, and the custom bit-multiplier scaling can be absorbed into the floating-point scaling operation (which is

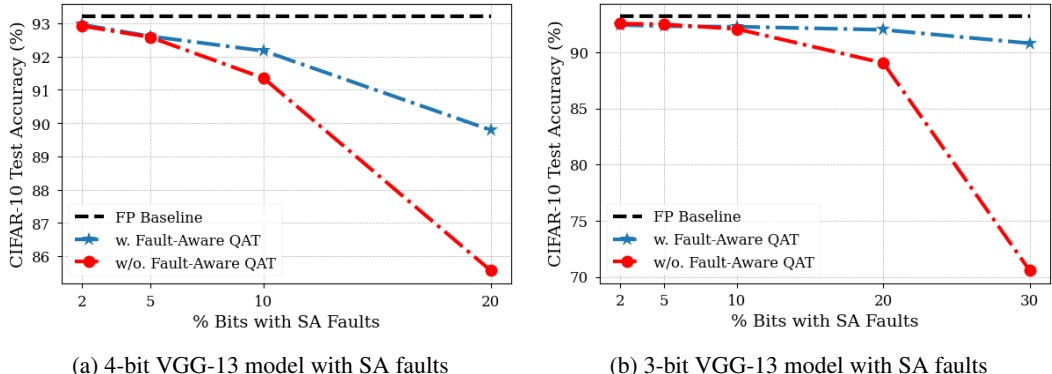

(a) 4-bit VGG-13 model with SA faults

(b) 3-bit VGG-13 model with SA faults

Figure 2: Performance preservation with SA faults: periodic faulty weight loading maintains accuracy for low fault densities; our fault-aware modified QAT extends robustness to high fault fractions.

common to all quantization schemes), thus eliminating any overhead caused by floating-point bit-multipliers. Learning custom bit multipliers within QAT enables highly effective low-bit quantization models that are compatible with standard in-memory computing architectures.

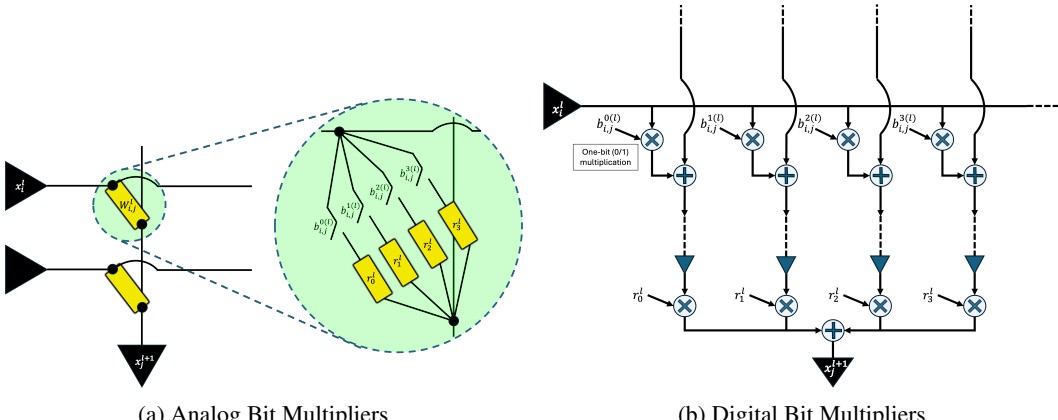

(a) Analog Bit Multipliers

(b) Digital Bit Multipliers

Figure 3: **(a)** Analog implementation. **(b)** Digital implementation. While uniform quantization uses bit-multipliers $(r_0^l, r_1^l, r_2^l, r_3^l)$ in powers-of-2 proportions $(1, 2, 4, 8)$, we propose learning custom multiplier factors instead.

## 5    Conclusion and Future Work

We introduce a novel algorithm for learning bit multipliers within QAT, enabling efficient low-bit quantization models with learnable, non-uniform levels compatible with in-memory computing architectures. Our approach demonstrates minimal accuracy drops for 3- and 4-bit models compared to FP baselines across various datasets and architectures, including CIFAR-10 and ImageNet using ResNet-18, and CIFAR10-DVS and N-Caltech 101 using spiking VGG-11 and ResNet-19. Notably, our quantized models occasionally outperform their FP counterparts. We further extend our method to address SA faults, maintaining performance with up to $30\%$ faulty bits. Future directions include extending the method to channel-specific quantizers, conducting fault-aware training experiments on additional benchmarks, expanding ANN and SNN model evaluations, and exploring sub-3-bit quantization. These advancements aim to enhance the efficiency and robustness of quantized neural networks for resource-constrained environments and hardware non-idealities.

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
