# OpenReview forum: "N Multipliers for N Bits: Learning Bit Multipliers for Non-Uniform Quantization"
_NeurIPS.cc/2024/Workshop/MLNCP — MLNCP Poster_

### Official Review · Reviewer_o438 · 2024-09-19
**To what precision are the $r$ values encoded?**

**Rating:** 5
**Confidence:** 3

**Review:**

The paper proposes a non uniform quantization method for deep nets by introducing layer wise parameters $r \in R^N$ and offset $c$ to model $N$ bit quantization. Essentially the weights are encoded as an affine product between the bits, $r$ and $c$.

The paper looks good upon quick reading but a question comes to mind after more careful thought: how are the parameters $r$, which are real valued (line 65), encoded in the hardware? To what precision? Sure the weight is encoded with a few bits, but the coefficients $r$ are real too and they are not powers of two, so it looks like the problem of quantizing the weights creates a new problem of quantizing the $r$ parameters... From a crossbar perspective like Fig 3a, it now means that one cell must not contain one resistive device but 4? This does not seem to be very practical for such devices where chip area is expensive.

I am not sure whether it qualifies as a fatal flaw in the reasoning. More explanation would be welcome concerning the circuits implementing the multiplier.

---

### Official Review · Reviewer_FfQo · 2024-10-04
**A non-uniform quantization method applicable for multiple neural network paradigms and target hardware architectures.**

**Rating:** 6
**Confidence:** 3

**Review:**

The submission introduces the author's method for a non-uniform quantization method applicable for ANNs, SNNs, and to address hardware faults that may arise in emerging ML accelerator technologies. To demonstrate the utility of their method, the authors present results across multiple neural networks (ResNet18, VGG11, ResNet19) and several datasets (CIFAR10, ImageNet, CIFAR10 DVS, N-Caltech101) at 4- and 3-bit precision.

The broad applicability of the method is a strength as many quantization methods are not necessarily applicable for ANNs as well as SNNs. A shortcoming of the submission is that as a consequence of their broad approach the narrative becomes difficult to follow as they jump from talking about ANNs to SNNs to hardware faults without transitions or a clear direction. For example, in the introduction the motivation about crossbar arrays and matrix-multiplications implies the manuscript may be hardware specific. However, they then go into text about quantization aware training only to then shortly thereafter go into spiking neuromorphic hardware examples. This pattern of briefly touching upon a topic to then move to another leaves the narrative disjoint and hard to follow.

 A critical shortcoming of the submission is how the author's approach compares with other methods. There is a large body of literature around quantization methods. How does this compare? The only representation of other methods is the results tables which present and reference other methods.

And in terms of the manuscript formatting, both the figures and tables could use improvements. Figure 1 b needs explanation as to what the red plot represents. The caption does not describe a or b. For enhanced clarity, recommend the tables are improved. The use of bold is very subtle (only bolding the delta and not the entire best performance value. And even the shorthand of W and A requires the reader realize that is pertaining to weights and activations.

Overall, the submission is well aligned with the goals of the MLNCP workshop. However, the shortcomings highlighted above are the basis for a marginally above acceptance threshold rating given there are key limitations to the work.

---

### Decision · Program_Chairs · 2024-10-10

Accept (Poster)